# The ASSIST trial: *A*cute effec*ts* of manipulating *s*trength exercise volume on *i*nsulin *s*ensi*t*ivity in obese adults: A protocol for a randomized controlled, crossover, clinical trial

**Luis Filipe Rocha Silva**[1], **Bruna Caroline Chaves Garcia**[2], **Elizabethe Adriana Esteves**[1,3], **Zachary Aaron Mang**[4], **Fabiano Trigueiro Amorim**[5], **Marco Fabrício Dias-Peixoto**[1,6], **Fernando Gripp**[6], **Valmor Tricoli**[7], **Flavio de Castro Magalhaes**[1,5,6]*

**1** Graduate Program in Health Sciences, Federal University of the Jequitinhonha and Mucuri Valleys, Diamantina, Minas Gerais, Brazil, **2** Laboratory of Exercise Biology and Immunometabolism, Centro Integrado de Pós-Graduação e Pesquisa em Saúde, Programa Multicêntrico de Pós-Graduação em Ciências Fisiológicas, Federal University of the Jequitinhonha and Mucuri Valleys, Diamantina, Minas Gerais, Brazil, **3** Department of Nutrition, Federal University of the Jequitinhonha and Mucuri Valleys, Diamantina, Minas Gerais, Brazil, **4** Post-doctoral Research Associate, Occupational Safety & Health, Los Alamos National Laboratory, Los Alamos, New Mexico, United States of America, **5** Department of Health, Exercise, and Sports Sciences, University of New Mexico, Albuquerque, New Mexico, United States of America, **6** Department of Physical Education, Federal University of the Jequitinhonha and Mucuri Valleys, Diamantina, Minas Gerais, Brazil, **7** School of Physical Education and Sport, University of Sao Paulo, Sao Paulo, Sao Paulo, Brazil

* fcm@unm.edu

## Abstract

Type 2 diabetes mellitus is a disease in which insulin action is impaired, and an acute bout of strength exercise can improve insulin sensitivity. Current guidelines for strength exercise prescription suggest that 8 to 30 sets could be performed, although it is not known how variations in exercise volume impact insulin sensitivity. Additionally, this means an almost 4-fold difference in time commitment, which might directly impact an individual's motivation and perceived capacity to exercise. This study will assess the acute effects of high- and low-volume strength exercise sessions on insulin sensitivity. After being thoroughly familiarized, 14 obese individuals of both sexes (>40 year old) will undergo 3 random experimental sessions, with a minimum 4-day washout period between them: a high-volume session (7 exercises, 3 sets per exercise, 21 total sets); a low-volume session (7 exercises, 1 set per exercise, 7 total sets); and a control session, where no exercise will be performed. Psychological assessments (feeling, enjoyment, and self-efficacy) will be performed after the sessions. All sessions will be held at night, and the next morning, an oral glucose tolerance test will be performed in a local laboratory, from which indexes of insulin sensitivity will be derived. We believe this study will aid in strength exercise prescription for individuals who claim not to have time to exercise or who perceive high-volume strength exercise intimidating to adhere to. This trial was prospectively registered (ReBEC #RBR-3vj5dc5 https://ensaiosclinicos.gov.br/rg/RBR-3vj5dc5).

**Data Availability Statement:** This is a protocol paper, so there is no data collected, and no means to share data.

**Funding:** This study is supported by the Conselho Nacional de Desenvolvimento Científico e Tecnológico (CNPQ: Grant#407975/2018-7 and #402091/2021-3) and by the Fundação de Amparo à Pesquisa do Estado de Minas Gerais (FAPEMIG: Grant#APQ-00008-22). The funders did not and will not have a role in study design, data collection and analysis, decision to publish, or preparation of the manuscript.

**Competing interests:** The authors have declared that no competing interests exist.

## Introduction

Diabetes Mellitus (DM) is a condition in which blood glucose levels remain high in the face of insufficient insulin production and/or reduced insulin sensitivity [1]. Hyperglycemia can lead to severe and life-threatening disorders, such as cardiovascular disease, nerve and kidney damage, amputation of limbs, vision loss and even blindness [1]. The number of people affected by DM has grown exponentially, and by 2017, this disease was the ninth leading cause of death in the entire world [2]. In 2021 it was responsible for 12.2% of global deaths in people aged 20–79 [1]. DM is related to high financial costs [3], and in Brazil, the total expenditure on DM in 2014 was more than R$ 26 billion (~US$ 5.5 bi) [4]. Approximately 90–95% of DM cases are type 2 (DM2) [5], which is related to the impairment of insulin sensitivity induced by obesity [6].

The beneficial effects of strength training on insulin sensitivity have been increasingly recognized. Interestingly, though, it has been reported that the training-induced improved insulin sensitivity may be lost after as few as 4–6 days following the last training session [7–9], suggesting the positive effects of exercise on glycemic control can be largely attributed to the acute improvements observed in the hours-days after each exercise bout [10]. Notably, several studies point to the beneficial effects of a single acute bout of strength exercise on improving insulin sensitivity [11], and it has been reported that strength exercise sessions increase insulin sensitivity for up to 48 hours [10]. Thus, understanding strength exercise prescriptions that elicits acute improvements in insulin action is of great clinical value.

Studies carried out over the last 30 years have shown that a strength exercise session is capable of improving insulin sensitivity. For instance, Koopman et al [12] reported that a strength exercise session (25 sets, 10 repetitions [reps], 75% 1 rep maximum [RM]) improved insulin sensitivity, measured by an insulin tolerance test, in healthy young men. Similar results were observed by Tong et al [13] who evaluated the effects a strength exercise session (9 sets, 10 reps, 75% 1RM) in nondiabetic men and showed a decreased glucose and insulin area under the curve (AUC) in response to an oral glucose tolerance test (OGTT). Andersen and Høstmark [14] observed in strength-trained men lower glycemic response to a meal test 14 hours after a session of strength exercises (21 sets, 10 reps, 65% 1RM), and more recently, Monroe et al [15] also studied trained men and reported improved capillary glucose levels after moderate- and high-load, volume-matched, strength exercise sessions (35 sets, 3 reps, at 90% 1RM and 15 sets, 9 reps, 70% 1RM). The acute benefits of strength exercise on insulin sensitivity have also been investigated in older and obese subjects, as well as patients living with prediabetes and DM2. For example, Bittel et al [16] demonstrated that a strength exercise session (21 sets, 10–12 reps, 80% 1RM) increased insulin sensitivity in response to a mixed meal in obese men, while Fluckey et al [17] evaluated young individuals and older patients living with and without DM2 in response to an acute strength exercise session (21 sets, 10 reps, 50, 75 and 100% of 75% 1RM) and observed a smaller insulin AUC in both young individuals and in DM2 patients. Van Dijk et al [10] studied individuals living with prediabetes, and with DM2 in response to a session of strength exercise (16 sets, 10 reps, 40–75% 1RM) and reported decreased average concentrations of glucose, and prevalence of hyperglycemia over 24 hours.

On the other hand, some studies did not demonstrate positive effects of a strength exercise session on insulin sensitivity. In apparently healthy individuals, one session of strength exercises (15 sets, 10 reps, 45–70% 1RM) led to unfavorable and clinically significant increases in insulin response [18], and in sedentary postmenopausal women no improvements were observed in insulin sensitivity, glycemia, and C-peptide response (an index of insulin release from beta cells) after a bout of strength exercise (21 sets, 10 reps, 50–100% 10RM) [19]. In strength trained men, Luebbers et al [20] reported no change in insulin sensitivity after high-

(24 sets, 8 reps, 85% 10RM) or low-load (24 sets, 15 reps, 45% 10RM) sessions of strength exercise. Furthermore, Malin et al [21] studied non-diabetic women with normal fat percentage (<35%) and with high fat percentage (>40%) and observed no change in insulin sensitivity after a bout of strength exercise (30 sets, 10–12 reps, 60% 1RM). Finally, no beneficial effects of a session of strength exercises (15 sets, 10 reps, 45–70% 1RM) on insulin sensitivity were observed individuals living with DM2 [22] or on glycemic control in a combined sample of individuals living with prediabetics and DM2 (18 sets, 10 reps, 65% 1RM) [22].

Clearly, the literature pertaining to the acute effects of a strength exercise session on insulin sensitivity is conflicting. However, it is possible to identify characteristics within exercise prescription variables that may explain the differing results. In a narrative literature review, Brown et al [23] identified 14 studies that evaluated the effects of a strength exercise session on insulin sensitivity. It was observed that concentric muscular failure (characterized as the inability to continue the set due to failure in the concentric moment of the movement of a given rep) is fundamental for observing improvements in glycemia and insulin sensitivity, since this certifies maximum (or close to maximum) effort during the sets. Moreover, it was noted that studies with positive results prescribed higher number of sets, which leads to higher exercise volume. In fact, a recent systematic review and meta-analysis reported that strength exercise sessions with 21 sets or more showed a greater improvement in glycemic control compared to sessions with fewer than 21 sets [24]. Nonetheless, current guidelines suggest that for improving glucose metabolism a strength exercise session could be composed of 8 to 10 exercises and 1 to 3 sets for each exercise [25], which translates into a wide range of total exercise volume and, by default, time commitment.

Studies that have assessed the acute effects of strength exercise volume on insulin sensitivity are scarce [11]. Reed et al [26] studied 1 or 3 sets (10 exercises, 10 reps, 65% 1 RM), performed in a circuit style program, in normoglycemic women, and found a significant improvement in insulin sensitivity only after the session with higher volume. Unfortunately, the prescription of strength exercise in circuit style prevents inferences for traditional strength exercise. Black et al [27] evaluated 8 exercises performed with 1 or 4 sets, either at 65% 1RM (12–15 reps) or 85% 1RM (6 to 8 reps). The authors observed improvements in insulin resistance in all protocols; but, protocols with single sets showed less effect. However, in that study, the degree of effort in each set was not reported and arguably higher in the high-volume conditions, and as this parameter is suggested as a determinant in the improvement of insulin sensitivity [23], one cannot exclude the possibility this factor played a role in the results observed. Furthermore, Black et al [27] assessed insulin resistance by the homeostasis model assessment of insulin resistance (HOMA-IR), which is more limited than other methods, such as OGTT-derived indexes [28], and more closely reflects hepatic insulin resistance [29]. These few studies in the literature, as well as their limitations, show the need for more research on the effects of strength exercise volume on insulin sensitivity.

Another important point regarding the prescription of strength exercise are the factors that prevent people from regularly engaging in exercise training. When asked about non-adherence to an exercise protocol, the most frequent justification is lack of time [30]. In addition, low self-efficacy (self-perception of the inability to accomplish something) is also one of the reasons people do not exercise frequently [31]. Additionally, positive feelings experienced in response to exercise are important factors for long-term adherence to an exercise program [32, 33]. Therefore, investigating the effect of low- and high-volume strength exercise–which leads to low and high time commitment, respectively [34]–on insulin sensitivity can be beneficial to motivate people who claim not to have time to perform strength exercises and for those who feel that they are not able to or do not experience positive feelings when performing high volumes of strength exercise.

## Objective

### Primary objective

The primary objective will be to investigate the acute effects of high- and low-volume strength exercise sessions on insulin sensitivity indexes in obese adults.

### Secondary objective

Assess the acute effects of high- and low-volume strength exercise sessions on feeling, enjoyment, and self-efficacy in obese adults.

## Methods

### Ethics approval

This study was approved by the local institutional review board (the Research Ethics Committee of the Federal University of the Jequitinhonha and Mucuri Valleys—certificate number CAAE 63190422.0.0000.5108) and complies with the Declaration of Helsinki. The present study was prospectively registered in a clinical trial registry (ReBEC #RBR-3vj5dc5 https://ensaiosclinicos.gov.br/rg/RBR-3vj5dc5). Any modifications to the protocol will be submitted to the ethics committee, followed by an updating of the trial registry.

### Description of participants

Altered insulin sensitivity is associated with increased lipotoxicity, a consequence of increased body fat [35]. Thus, the inclusion criteria are obese individuals of both sexes, aged over 40 years, with central obesity (waist circumference > 102 cm in men and > 88 cm in women), with stable body mass (±3 kg) in the last 3 months and able to perform physical activity [36]. To confirm obesity and central obesity statuses of the participants, dual-energy X-ray absorptiometry will be used to assess fat percentage and visceral fat mass (see details below in **Anthropometric measurements**), and participants will be included only if both indexes are >90th percentile for age and sex [37]. Exclusion criteria are individuals with signs, symptoms or presence of diabetes or any other metabolic disease, cardiovascular disease, cerebrovascular disease, kidney disease, respiratory disease, and osteoarticular disease [36]. In addition, those who report using any medication that may influence the expected results (including oral contraceptives [38]) and use of anabolic steroids or being pregnant will be excluded. Furthermore, participants who describe dietary supplement intake known to affect exercise performance, such as caffeine, beta-alanine, creatine, and sodium bicarbonate [39] will be excluded. Participants will be from the local community (Diamantina, MG–Brazil) and the project will be publicized through social media, posters at strategic points in the city, and word of mouth. After showing interest, participants will receive information by the researchers (LFRS or FCM) about the objectives of the work, possible discomforts, risks and benefits before volunteering. After being thoroughly informed of the study's risks and benefits, and before initiating participation, subjects will sign an "Informed Consent Form" (available upon request), previously approved by the local institutional review board. Participants will be reimbursed for expenses incurred due to the research and resulting from it. At any time if they suffer any damage, unarguably resulting from this research, they will be entitled to compensation, full and immediate assistance, free of charge for as long as necessary.

### Study type and design

This study will be a randomized controlled, crossover, 3-arm, clinical trial. The protocol description followed the SPIRIT [40] and the TIDieR [41] guidelines (Fig 1 and S1–S3 Tables).

| | STUDY PERIOD | | | | | | | |
|---|---|---|---|---|---|---|---|---|
| | Enrolment | Before randomization | Allocation | Interventions | | | Post Intervention | |
| TIMEPOINT** | -t₁ | | 0 | Day 1 | Day 8 | Day 15 | +30 min | +10-11 hours |
| **ENROLMENT:** | | | | | | | | |
| **Eligibility screen** | X | | | | | | | |
| **Informed consent** | X | | | | | | | |
| **Allocation** | | | X | | | | | |
| **INTERVENTIONS\*:** | | | | | | | | |
| *High-volume session* | | | | ←—→ | | | | |
| *Low-volume session* | | | | | ←—→ | | | |
| *Control session* | | | | | | ←—→ | | |
| **ASSESSMENTS:** | | | | | | | | |
| *Anthropometrics* | | X | | | | | | |
| *Familiarization* | | X | | | | | | |
| *Strength test* | | X | | | | | | |
| *Subjective measurements* | | | | | | | X | |
| *OGTT* | | | | | | | | X |

**Fig 1. SPIRIT schedule of assessments at different time points.** OGTT: oral glucose tolerance test; *Sessions order is randomized with >3-day washout period.

Participants will undergo a health history questionnaire, anthropometric measures which include height, body mass, waist circumference and body composition, a period of familiarization and strength assessment. At least 4 days after the strength tests, they will carry out 3 randomized experimental sessions, separated by at least 4 (but no more than 28) washout days, as it has been shown that the acute effect of exercise on improved insulin action lasts up to 2 days [42]. In case the sessions are scheduled 21 to 28 days apart, participants will be required to perform a familiarization session 2 weeks after the preceding (and 1 to 2 weeks before the following) experimental session to avoid defamiliarization. The interval between experimental sessions will be recorded and reported in the final manuscript, and discussed accordingly. To avoid the variation in insulin sensitivity across the menstrual cycle [43–49], in case premenopausal women are included in the sample, they will be assessed in the follicular phase (days 1–14) [9, 26, 50–52] of their menstrual cycle. Two of the experimental sessions will be performed with strength exercises: a high-volume session [3 sets per exercise–high volume]; a low-volume session [1 set per exercise–low volume]; and a control session: no exercise. In the exercise sessions, 7 exercises will be performed for the major muscle groups (1 –hex bar squat; 2 –bench press; 3 –leg press; 4 –lat pulldown; 5 –leg extension; 6 –shoulder press; 7 –leg curl; Fig 2), totaling 21 sets in the high-volume session and 7 sets in the low volume session [24]. The sessions will be held at night, between 8:00 pm and 9:00 pm. This time schedule for the sessions is ecologically valid, as exercise in the evening is commonly observed in fitness centers due to time constraints for exercising earlier during the day [13, 53], and it has been shown

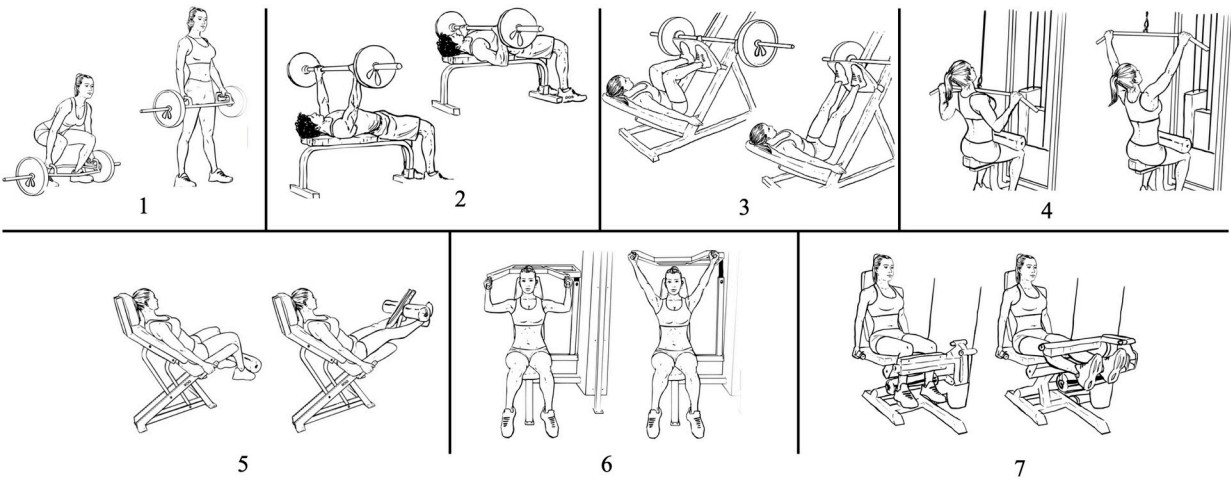

**Fig 2. Exercises that will be prescribed for the strength exercise sessions.** 1. Hex bar squat, 2. Bench press, 3. Leg press, 4. Lat pulldown, 5. Leg extension, 6. Shoulder press, 7. Leg curl.

that night-time exercise does not affect sleep duration or quality [53]. The next morning between 7:00 am and 8:00 am (between 10 and 11 hours after the session), an OGTT will be performed in a local clinical laboratory. Participants will respond to scales of feeling, enjoyment, and self-efficacy after the exercise sessions. All sessions will be supervised by a certified fitness professional. The experimental design is illustrated in Fig 3.

Subjects' participation will be discontinued in case of withdrawn consent, or in case the interval between sessions is higher than 28 days. We will report reasons for withdrawal and discuss the reasons qualitatively in the final manuscript. Adverse events related or nonrelated to the research protocol experienced by participants during study (e.g. skeletomuscular injury, having a cold, etc.) will be carefully recorded and reported in the final manuscript.

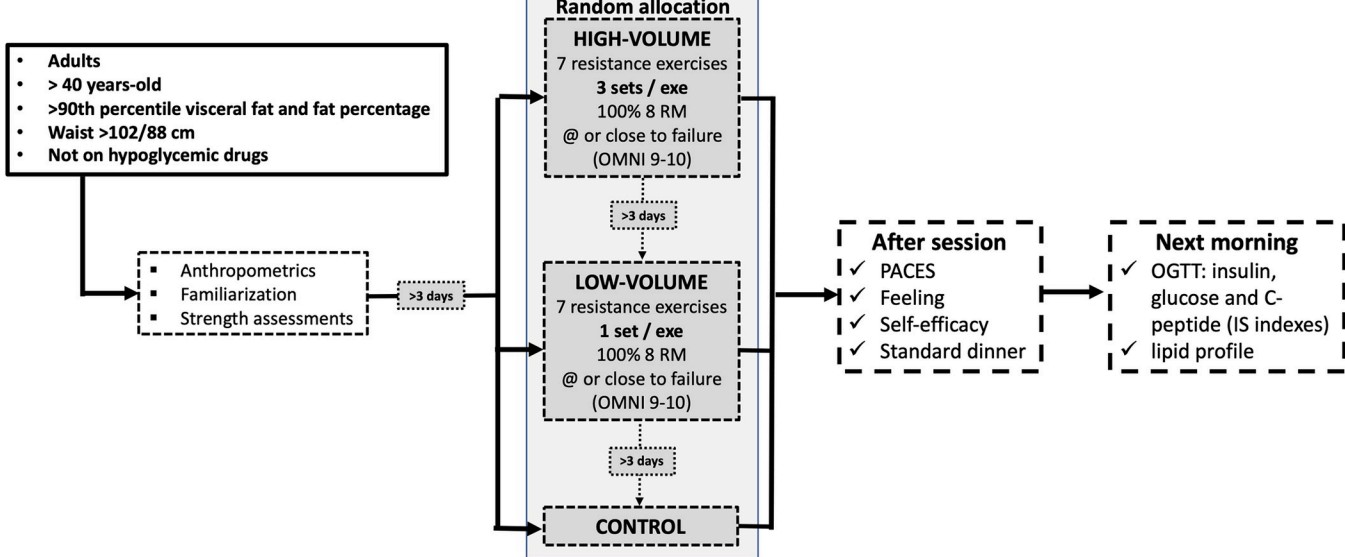

**Fig 3. Illustration of the experimental protocol timeline.** BMI: body mass index; PACES: physical activity enjoyment scale; OGTT: oral glucose tolerance test; IS: insulin sensitivity.

### Pre-participation screening

Participants will answer a health questionnaire based on the most recent recommendations of the American College of Sports Medicine [36] that will assess possible risks of performing the exercise protocol.

### Anthropometric measurements

Individuals will be submitted to the measurement of body mass and height on an analog scale, with a stadiometer attached, to calculate the BMI. Subsequently, the waist circumference of the participants will be measured. Then, the body composition will be evaluated by means of dual-energy X-ray absorptiometry (DXA, Lunar, iDXA Advanced). Participants will be accommodated in the device, following the manufacturer's instructions for analysis of fat mass, fat-free mass, and visceral fat mass.

### Familiarization

A thorough familiarization period will be carried out in each of the 7 strength exercises (3 sets/exercise x 8 reps/set). On the first day, the participant will be instructed to lift a light load, in which performing 8 reps is considered "somewhat easy" or between "3" and "4" according to the OMNI-RES scale [54]. In the second session, a load considered "somewhat hard" or between "5" and "7" should be lifted. In the third session, the load will be considered "hard", or between "7" and "9". In the 4th session, they should lift loads between "hard" and "extremely hard" or between "9" and "10", mimicking the effort expected in the strength test itself. The interval between sets will be 120 seconds. After this, a fifth familiarization session will be performed to mimic the strength tests (see below in **Strength tests**). In all sessions, the correct form for each exercise (range of motion, duration of concentric:eccentric phases [~1:2 sec; aka tempo]) will be carefully observed, and corrections will be made. Furthermore, in every session, participants will be familiarized with the subjective measurements (feeling, enjoyment, and self-efficacy, see below). All sessions will be supervised by a certified fitness professional. The interval between familiarization days will be at least 48 but no more than 168 hours (2 to 7 days).

### Strength tests

At least 72 hours after the last familiarization session, participants will perform the 8-RM test on each of the 7 exercises. After an initial 5 minute warm-up walking on a treadmill (~3–4 km/h), and 1 set with low-load (12 reps, ~40–50% 1RM, ~3–4 OMNI), the participant will be instructed to perform 8 reps, and the load will be gradually increased with each set until the participant is able to perform only 8 reps (with full range of motion and tempo ~1:2 sec). The rest period between attempts will be between 120 and 180 seconds [55]. Because during the high- and low-volume experimental sessions participants will perform all seven exercises sequentially, and fatigue will likely accumulate as the sessions progresses, we decided to have their 8-RM tested in the same fashion, in order to mimic their anticipated effort. This means a true 8-RM might not be recorded, especially for exercises tested later in the strength test session. However, we believe this will not interfere with our study design and results interpretation, as exactly the same load will be prescribed for the exercise experimental sessions. All tests will be performed under the supervision of the same person, and the results will serve as a basis for prescribing exercise sessions.

### Random allocation

The order in which the experimental sessions will be held will be randomized with the aid of the website https://www.randomizer.org. The random orders generated by the program will

be printed and inserted inside opaque, sequentially numbered envelopes, which will later be sealed by a research collaborator not directly involved in data collection. After completing the strength tests, the envelope will be opened, and the sequence of experimental sessions that will be followed will be revealed.

## Blinding procedures

Due to characteristics inherent to this trial (exercise), blinding of participants and researchers involved in data collection is not possible. However, employees from the local clinical laboratory responsible for blood harvesting, processing, and analysis will be blinded to allocation. For that, participants will be instructed not to reveal details about the protocol to the clinical laboratory personnel. Moreover, the researcher responsible for data analysis will remain blind to allocation until completion of analysis (see below in *Data analysis*).

## Dietary control

Participants will be instructed by a certified nutritionist on how to log their diet in a notebook on 3 nonconsecutive days of the week, one of the days being on the weekend [56]. Daily caloric intake in kilocalories (kcals) and kcals consumed from lipids, protein and carbohydrates will be analyzed using nutritional software (Dietbox Software, Brazil). Thereafter, a food plan will be given to the participants who will follow exactly the same food plan the day before and the day of the experimental sessions. This food plan will not change the composition and amounts of the daily basis food consumption of the participants, but they will be instructed to consume the same meals and the same amounts of food at the same hours of the day during the experimental protocol days, including a pre-session meal composed of 65% carbohydrates, 15% proteins, and 20% lipids one hour before reporting to the gym (~6:30 pm). Frequent text messages will be sent to participants to ensure they followed instructions. Adherence to dietary instructions will be verified by assessing their food record. After the experimental sessions, a standard snack (65% carbohydrates, 15% proteins, and 20% lipids) will be offered, which will be ingested between 9:00 and 9:30 pm at the gym where the sessions will take place, with calories determined from the individual daily needs, plus the estimated net energy expenditure of the session [57]. Participants will be instructed not to consume food after the offered snack and must fast until the morning of the next day to perform the OGTT in a local laboratory.

## Interventions

Participants will be instructed not to perform moderate/high physical activity during the 48 hours that precede the experimental sessions. Frequent text messages will be sent to participants to ensure they follow instructions. Adherence to physical activity instructions will be verified before each session by asking the participants.

**High-volume session.** All sessions will be held at the university facilities and will always be accompanied by the same certified fitness professional. Participants will report to the gym at 7:30 pm, where they will remain at rest until 8:00 pm, only receiving guidance on the exercises to be performed and will respond to the pre-session affection scale (see below in **Subjective measurements**), as suggested by Alves et al [58]. The experimental session will start at 8:00 pm and will be as follows: an initial 5 minute warm-up walking on a treadmill (~3–4 km/h) followed by 7 strength exercises that recruit the major muscle groups, performed in the order presented below: 1 –hex bar squat; 2 –bench press; 3 –leg press; 4 –lat pulldown; 5 –leg extension; 6 –shoulder press; 7 –leg curl. Participants will perform 3 sets with the previously determined 8-RM load, with as many reps per set as they tolerate (until concentric muscular failure–determined as the inability to maintain full range of motion or the inability to maintain

tempo (~1:2 sec) for 2 consecutive reps, or by voluntary fatigue of the participant), with a rest of 120 s between sets and exercises. At the end of each set, they will respond to the OMNI-RES scale to ensure that they have reached maximum effort (9 or 10 on the scale). The number of reps performed in each set will be recorded to calculate the total exercise volume-load [59]. Based on pilot studies, sometimes the participant might not be able to complete the first rep of a given set, especially for latter exercises (e.g. leg extension and leg curl). In case that happens, we will not consider that set, will reduce the load by 5–10%, and give them 1 minute to rest after which another attempt will be made. If even after this the participant is unable to complete the first rep of the set, we will consider that the number of reps for that set was 0, and will move on to the next exercise, or finish the session (if this happens during the last exercise, e.g., leg curl). Also based on pilot studies, we anticipate this session will end between 8:45 and 8:55 pm. After that, participants will remain sedentary, answer the feeling, enjoyment, and self-efficacy scales (see below in **Subjective measurements**), and ingest the standard meal described above between 9:00 and 9:30 pm. After an overnight fast, they will report to a clinical analysis laboratory the next day in the morning (between 07:00 and 08:00 am) to perform the OGTT. Thus, the interval between the end of the session and the OGTT will be ~10 to 11 hours.

**Low-volume session.** In this session, all procedures will be identical to the high-volume session, with the exception of the number of sets performed; in this case, only 1 set will be performed in each of the 7 strength exercises. This session will last ~15–20 min. To avoid variations in the time between the end of the session and the OGTT, this session will start between 8:35–8:40 pm so that it ends at the same time as the high-volume session.

**Control session.** In this session, all procedures (including pre-session instructions) will be identical to the high-volume session, with the exception of performing the strength exercises. However, to simulate all other procedures performed in the high-volume session, participants will follow all instructions and, in the gym, will wear the same clothes, sit on the equipment for the same amount of time as the high-volume session, without any reps being performed. Subjective measures and OMNI will also be responded by the participants mimicking the high-volume session.

## Subjective measurements

To verify feeling, the scale described by Hardy and Rejeski [60] and validated to Portuguese by Alves et al [58] will be used. This scale will be answered by the participants before and after the experimental session [58]. To assess enjoyment, the physical activity enjoyment scale (PACES) described by Kendzierski and DeCarlo [61] and validated in Portuguese by Alves et al [58] will be used. To assess self-efficacy, a modified scale proposed by McAuley et al [62] will be used.

## Primary blood outcomes–OGTT-derived indexes

After ~9–10 hours of overnight fasting and ~10–11 hours after each experimental session, participants will report to a local clinical laboratory, and a blood sample will be collected (minute 0). Then, they will ingest 75 grams of glucose in 300 ml of water, and blood samples will be collected every 30 minutes until 120 minutes after glucose ingestion, totaling 5 withdrawals (minute 0, minute 30, minute 60, minute 90, and minute 120). Plasma concentrations of glucose, insulin, and C-peptide (a marker of insulin production) will be analyzed in each sample. With the insulin and glucose results at minute 0 (fasting values), insulin resistance will be calculated using HOMA-IR with the formula glucose (mmol) x insulin (μU/ mL) ÷ 22.5 [63], and insulin sensitivity using the quantitative insulin sensitivity check index [64]. Several insulin sensitivity indexes will be derived from the OGTT data [28], such as the oral glucose insulin sensitivity index [65], the Matsuda insulin sensitivity index [66] and Cederholm's index [67], muscle

insulin sensitivity index [68, 69], glucose-stimulated insulin sensitivity index [21], Gutt index [70], Avignon et al. index [71], Belfiore et al. index [72], Stumvoll et al. index [73], and McAuley et al. index [74], the simple index of insulin sensitivity [75], and as an index of hepatic insulin resistance, the hepatic insulin resistance index [68]. In addition, the glucose, insulin, and C-peptide AUC will be calculated using the trapezoidal method [76].

## Secondary blood outcomes

**Lipid profile.** Fasting blood samples will be analyzed for the lipid profile (total cholesterol and fractions and triglycerides).

## Statistics

**Sample size.** The sample size was calculated from the literature [10, 12, 16, 17, 19, 23, 26, 27, 77] using the G*Power program (Heinrich-Heine-Universität Düsseldorf, Germany, version 3.1.9.6), inserting the parameters for a one-way ANOVA, effect size of 0.6 probability of error type alpha of 0.05 and power (probability of error type 1 –beta) of 0.9. With these parameters, 14 individuals will be needed (actual power of 0.93). Due to the usual sample loss of 25–40%, we will initially recruit 20 participants. In order to promote participant retention, frequent text messages will be sent as reminders to follow the instructions, and to show up in the scheduled sessions. Also, in case the participant misses a session, all effort will be made to reschedule as long as it is within 28 days of the previous session. Dropout reason will be recorded and reported in the manuscript.

**Data analysis.** To ensure confidentiality, all study-related information will be stored securely at the study site. All participant information will be stored in locked file cabinets in areas with access controlled by the principal investigator (FCM). All laboratory specimens, reports, data collection, process, and administrative forms will be identified by a coded ID number only to maintain participant confidentiality. All records that contain names or other personal identifiers, such as locator forms and informed consent forms, will be stored separately from study records identified by code number. No personal information from participants will be released outside the study. Full access to the complete dataset will be permitted only to LFRS and to FCM.

The statistical analysis will be conducted following the principles of intention-to-treat analysis [78]. Thus, data from subjects will be assessed as randomized, regardless of whether they received the randomized treatment, meaning that even if they fail to follow the physical activity and diet pre- and post-session instructions, or fail to complete the sessions as required (e.g. fail to achieve 9–10 in the OMNI scale), or fail to perform the OGTT at the schedule time, so-called non-adheres, their data will be included in the analyses. Furthermore, we will analyze the data "per protocol" [79], meaning that participants who deviate from instructions will be excluded from final analyses. Differences in results stemming from intention-to-treat and per protocol analyses will be discussed accordingly. The available variables will also be compared between participants who withdraw from the study and those who remain. Incomplete datasets (e.g., participant who complete 1 or 2 of the 3 sessions) will analyzed via sensitivity analysis of augmented data sets. Dropouts will be included in the analysis by modern imputation methods for missing data [80].

Data will be expressed as the mean and standard deviation with a confidence interval of 95%. For the analysis of data normality, we will perform the Shapiro-Wilk test. For normally distributed data, analysis of variance will be used with one source of variation (experimental situation). If a significant main effect was observed, post hoc Tukey test was used. For non-parametric data, the Kruskal-Wallis test was used, or Friedman's test, when necessary. Effect

size was calculated and interpreted as follows: <0.2 = no effect, 0.2–0.49 = low effect, 0.5–0.79 = medium effect, and greater than 0.8 = large effect [81, 82]. The significance level will be 5%. The Prisma program (GraphPad Software, San Diego, CA-USA–version 8.4.0) will be used to analyze the results. The trial will be reported following the CONSORT guidelines [83, 84].

As stated above under **Blinding procedures**, the researcher responsible for data analysis (the statistician) will be blinded for group allocation. For that, a collaborator outside the research team will double enter data coded for allocation into a computer in separate spreadsheets so that the researcher responsible for data analysis can assess data without having access to information about the allocation. Allocation will only be revealed after statistics are run. In case there are doubts about data entry (missing values, or values outside the reference range), the researcher responsible for data analysis will contact the collaborator outside the research team responsible for double entry of the data to check accuracy. There will be no interim analysis.

## Publishing policy

The results stemming from this trial will be published in journals of the field regardless of the magnitude or direction of the outcomes.

## Authorship eligibility

Substantive contributions to the design, conduct, interpretation, and reporting of this trial will be recognized through the granting of authorship on the final manuscript stemming from this trial.

## Discussion

### Potential impact and significance of the study

An acute bout of strength exercise has been shown to improve insulin action and glucose metabolism [23]. However, current guidelines for the prescription of strength exercise suggest that 8 to 30 sets could be performed [25], which means a ~4-fold difference in exercise volume and thus time commitment. Furthermore, high-volume strength exercise might discourage individuals with low self-efficacy, which might diminish long-term adherence to a training program. Therefore, assessing the effects of low- vs high-volume strength exercise on insulin action and glucose metabolism bears clinical relevance because if low-volume strength exercise elicits improvement in these parameters, individuals might be more motivated and feel more confident in performing low-volume strength exercise.

### Strengths and weaknesses

**Strengths.**   This study will be a randomized clinical trial. Additionally, to reduce the risk of bias, the researcher who will perform the statistics will be blinded to the treatment. Furthermore, we will employ principles of intention-to-treat analysis [78], and we will assess the characteristics of those participants who do not complete the trial and compare them to those who do complete the trial. The strength exercises selection was based on studies that suggested using least 21 sets per session [24], performing multi-joint exercises that recruit large muscle mass, carrying the sets to or close to concentric muscular failure, and allowing 120 seconds recovery between sets and exercises [23]. Additionally, we will have a prolonged period of familiarization to ensure that participants are performing the exercises correctly, and a certified fitness professional will be present at every session. Finally, we will ask participants to

follow the diet prescribed by a certified nutritionist and will verify whether participants followed the prescribed diet by assessing their food logs. Moreover, we will provide a standard, individualized, energy expenditure-adjusted meal after the sessions to ensure that energy and carbohydrate replenishment are not confounding factors on post-exercise insulin sensitivity [85, 86]. Finally, although the OGTT might not be the gold-standard tool to assess insulin sensitivity compared to the euglycemic-hyperinsulinemic clamp (EHC), ingesting 75 grams of glucose is considered to be a more physiological stimulus compared to the supraphysiological insulin levels during EHC [87]. Moreover, the insulin sensitivity indexes that will be calculated from OGTT in the present study show high correlation (r = 0.61 and 0.96) [72] with EHC results. Also, OGTT is considered reliable and consistent for estimating insulin sensitivity over consecutive days [88]. Last, the majority of studies that assessed the acute effects of a strength exercise session on insulin sensitivity have employed OGTT as the assessment tool [9, 10, 13, 17–19, 26, 50, 77, 89–91].

**Weaknesses.** This study cannot be completed in a double-blind manner, as the participants and the researchers who will be present during the sessions will be aware of the treatment (high- or low-volume or control session). However, the professionals at the clinical lab who will collect and analyze the blood will be blinded to treatment, as well as the researcher who will perform the statistics. We will include obese men and women over 40 years old, so our results might not be readily translated to other clinical populations, such as younger, healthy individuals or individuals living with DM2. Furthermore, individuals with potentially different levels of insulin resistance might be included in this study, and greater exercise-induced improvement in insulin resistance has been observed in individuals with higher levels of baseline insulin resistance [26, 92]. Thus, this factor can also increase results variability. However, it should be pointed out that studying overweight/obese subjects regardless of their insulin resistance levels or fasting glucose impairment status is commonplace in the related literature [8, 93–95]. Nevertheless, the fact that participants will be their own control minimizes this potential source of variability. Moreover, individual results will be presented and discussed accordingly should individual levels of insulin resistance affect average group results. Furthermore, DM2 patients will be excluded, following our exclusion criteria. Also important to point out, participants might not follow the dietary and physical activity instructions the day before and the day of the sessions, which might also be confounding factors when interpreting our results. However, we will stress the importance of following these instructions by sending frequent text messages, and adherence will be verified when they report to the gym, and we will ask them to report what they ate to the nutritionist, which we believe increases adherence to diet prescription. Finally, per our study design, we will assess two significantly different strength exercise volumes, and by default time commitment, as lack of time is usually listed as one of the main reasons not to adhere to exercise [30]. Thus, our study design is not intended to explore a potential dose-response relationship between strength exercise volume and improvements in insulin sensitivity, although this approach would bear practical implications.

## Supporting information

**S1 Table. SPIRIT 2013 checklist.**
(DOCX)

**S2 Table. WHO trial registration data set.**
(DOCX)

**S3 Table. The TIDieR (Template for Intervention Description and Replication) checklist.**
(DOCX)

**S1 File.**
(DOCX)

**S2 File.**
(DOCX)

## Acknowledgments

This study will be based at the Federal University of the Jequitinhonha and Mucuri Valleys (Diamantina-MG, Brazil) which will provide the equipment and space (DXA, strength training room, strength training equipment, etc.) necessary for conducting the research.

## Author Contributions

**Conceptualization:** Luis Filipe Rocha Silva, Bruna Caroline Chaves Garcia, Elizabethe Adriana Esteves, Zachary Aaron Mang, Valmor Tricoli, Flavio de Castro Magalhaes.

**Funding acquisition:** Flavio de Castro Magalhaes.

**Methodology:** Luis Filipe Rocha Silva, Bruna Caroline Chaves Garcia, Elizabethe Adriana Esteves, Zachary Aaron Mang, Fabiano Trigueiro Amorim, Flavio de Castro Magalhaes.

**Writing – original draft:** Luis Filipe Rocha Silva, Bruna Caroline Chaves Garcia, Zachary Aaron Mang, Valmor Tricoli, Flavio de Castro Magalhaes.

**Writing – review & editing:** Elizabethe Adriana Esteves, Fabiano Trigueiro Amorim, Marco Fabrício Dias-Peixoto, Fernando Gripp, Valmor Tricoli, Flavio de Castro Magalhaes.

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
