## [Decision Letter · Decision Letter 0]

7 Dec 2023

PONE-D-23-04520

The ASSIST trial: Acute effectS of manipulating Strength exercise volume on Insulin SensiTivity in obese adults: a protocol for a randomized controlled, crossover, clinical trial

PLOS ONE

Dear Dr. Magalhaes,

Thank you for submitting your manuscript to PLOS ONE. After careful consideration, we have decided that your manuscript does not meet our criteria for publication and must therefore be rejected.

I am very disappointed to inform you that the Reviewers and the editorial team have decided to reject your manuscript.

As I have already informed you, I have taken charge of your manuscript after the unavailability of the last editor and I have done my best to complete the evaluation process as quickly as possible.

Kind regards,

Dured Dardari, Ph.D

Academic Editor

PLOS ONE

Reviewer's Responses to Questions

**Comments to the Author**

1. Does the manuscript provide a valid rationale for the proposed study, with clearly identified and justified research questions?

Reviewer #1: Yes

Reviewer #2: Partly

2. Is the protocol technically sound and planned in a manner that will lead to a meaningful outcome and allow testing the stated hypotheses?

Reviewer #1: Yes

Reviewer #2: Partly

3. Is the methodology feasible and described in sufficient detail to allow the work to be replicable?

Reviewer #1: Yes

Reviewer #2: Yes

4. Have the authors described where all data underlying the findings will be made available when the study is complete?

Reviewer #1: Yes

Reviewer #2: Yes

5. Is the manuscript presented in an intelligible fashion and written in standard English?

Reviewer #1: Yes

Reviewer #2: Yes

6. Review Comments to the Author

You may also provide optional suggestions and comments to authors that they might find helpful in planning their study.

Reviewer #1: The reviewed paper is a description of the protocol of a planned study, aiming to assess strength exercise volume on insulin sensitivity.

It seems to be an interesting approach that may add additional information to the existing evidence regarding positive influence of exercise on the metabolism. I would suggest only one measure that the authors could ask to the protocol, that is control but also use of heart rate during and after exercise as an additional parameter defining exercise. It may be that the same exercise may have different metabolic effect depending on how trained is a subject or patient.

The other issue are the inclusion criteria. I would suggest to assess at least some laboratory parameters, like fasting plasma glucose(twice), OGGT , HbA1c and lipids – to answer whether participants have prediabetes, and to exclude diabetes (which is quite often undiagnosed in obese subjects) or high lipids. Maybe the authors (names and specialties have been blinded to me) would wish to consult a diabetologist/endocrinologist?

Additionally, please justify the decision to perform the session between 8 and 9 pm the day before OGTT – why not earlier or later?

Minor issues: explaining that insulin sensitivity is an opposite to insulin resistance does not seem necessary to me. Please explain every abbreviation after its first appearance – like BRL, RM or others. Please define the time interval between the last familiarization session and study start.

Reviewer #2: Due to the inconclusive results of previous studies, the problem of the impact of particular strength training on insulin resistance remains unresolved.

However, the research protocol raises my concerns:

– Too broad inclusion criteria for the study. It is observed in the clinical practice that an obese person may do not have insulin resistance. Moreover, the BMI criterion is insufficient because healthy, physically active people with developed muscle tissue may have a high BMI, but this does not mean that they are obese from a biochemical point of view. In my opinion, the study group should comprise people with confirmed laboratory indicators of insulin resistance, e.g. based on HOMA-IR.

– Measuring insulin resistance based on the OGTT result seems to represent imprecise methodology. Basing basically the entire study on this parameter seems to be an oversimplification and lead to false conclusions.

– Performing a glucose tolerance test several hours after exercise seems inappropriate, because "acute" processes related to glucose utilization by the tissues are still taking place at that time. Thus, the determined parameters will reflect the body's acute response to increased glucose utilization rather than the possible increase in insulin sensitivity. It seems reasonable to increase the time interval between the training session and blood drawing and measure the true change in insulin sensitivity,

– Small number of participants in the study. Such sample size does not ensure that analyzed parameters will reach adequate power. In the power analysis of the test only the Matsuda insulin sensitivity index was used.

– Presented research program is not innovative.

The subject matter of the work is more consistent with journals on physiotherapy or exercise physiology.

7. PLOS authors have the option to publish the peer review history of their article (what does this mean?). If published, this will include your full peer review and any attached files.

Reviewer #1: No

Reviewer #2: No

- - - - -

---

## [Author Response · Author response to Decision Letter 0]

2 Jan 2024

We thank the reviewers for their time and effort on reviewing the protocol and provide below a point-by-point response to each of the comments in bold letter.

Reviewer #1: The reviewed paper is a description of the protocol of a planned study, aiming to assess strength exercise volume on insulin sensitivity.

It seems to be an interesting approach that may add additional information to the existing evidence regarding positive influence of exercise on the metabolism. 

We thank the reviewer for the comment.

I would suggest only one measure that the authors could ask to the protocol, that is control but also use of heart rate during and after exercise as an additional parameter defining exercise. It may be that the same exercise may have different metabolic effect depending on how trained is a subject or patient.

We appreciate this suggestion. However, resistance exercise intensity is not traditionally controlled by heart rate, nor is this response commonly assessed in resistance exercise studies. Also, in the present protocol we are interested in the metabolic, insulin-sensitizing effects of resistance exercise, not in the cardiovascular responses. Thus, we believe that adding this measurement will not add much to the protocol, or contribute to our aims, and may in fact burden the research personnel and participants. Therefore, we decided not to include heart rate measurement in this protocol. Nevertheless, if the reviewer strongly feels there is a rationale for this assessment, and thinks it would improve the quality of the protocol, we can consider including it.

The other issue are the inclusion criteria. I would suggest to assess at least some laboratory parameters, like fasting plasma glucose(twice), OGGT , HbA1c and lipids – to answer whether participants have prediabetes, and to exclude diabetes (which is quite often undiagnosed in obese subjects) or high lipids. Maybe the authors (names and specialties have been blinded to me) would wish to consult a diabetologist/endocrinologist?

We thank the reviewer for raising this important point and allowing us to address it. Type 2 diabetes mellitus (DM2) patients will be excluded from the study. Although we are not running pre-enrolment glucose assessments, in case DM2 is evident from the first OGTT they perform, participant will be excluded, following the reviewer’s suggestion and our exclusion criteria.

Additionally, please justify the decision to perform the session between 8 and 9 pm the day before OGTT – why not earlier or later?

Night-time exercise is common practice is “real-life” settings (e.g., fitness/health centers) as time constraints in modern life might hinder exercising earlier in the day. Also, studies on the effect of resistance exercise on insulin sensitivity have been conducted at this time [1]. Furthermore, data from a recent systematic review and meta-analysis showed no effect of night-time exercise on sleep quality or duration [2], suggesting night-time exercise can and should be performed. We added the following sentence in the manuscript under “Study type and design” to address the scheduling of our exercise sessions:

“This time schedule for the sessions is ecologically valid, as exercise in the evening is commonly observed in fitness centers due to time constraints for exercising earlier during the day [54,55], and it has been shown that night-time exercise does not affect sleep duration or quality [54].”

Minor issues: explaining that insulin sensitivity is an opposite to insulin resistance does not seem necessary to me. 

We have removed this from the abstract and introduction. We thank the reviewer for this suggestion.

Please explain every abbreviation after its first appearance – like BRL, RM or others. 

Abbreviations were defined accordingly.

Please define the time interval between the last familiarization session and study start.

Under “Study type and design” we explain the timeline of the study: 

“After at least 4 days, they will carry out 3 experimental sessions, performed at random, separated by at least 4 (but no more than 28) washout days, […].”

Also, in figure 3 we highlight the interval between end of familiarization and beginning of the first session.

 

Reviewer #2: Due to the inconclusive results of previous studies, the problem of the impact of particular strength training on insulin resistance remains unresolved.

We thank the reviewer for this comment, but it is extremally important we clearly establish that the current study relates to the effects of an acute bout of resistance exercise, not training (several consecutive bouts over weeks/months/year). This distinction will set the scenario for addressing what appears to be one of the main points raised by the reviewer. To further clarify this point, we added to the “Introduction” the following sentence:

“Interestingly, it has been reported that the training-induced improved on insulin sensitivity may be lost after as few as 4-6 days following the last training session [7,8], suggesting the positive effects of exercise on glycemic control can be largely attributed to the acute improvements observed in the hours-days after each exercise bout [9]. Notably, several studies point to the beneficial effects of a single acute bout of strength exercise on improving insulin sensitivity [10], and it has been reported that strength exercise sessions increase insulin sensitivity for up to 48 hours [11].”

However, the research protocol raises my concerns:

– Too broad inclusion criteria for the study. It is observed in the clinical practice that an obese person may do not have insulin resistance. Moreover, the BMI criterion is insufficient because healthy, physically active people with developed muscle tissue may have a high BMI, but this does not mean that they are obese from a biochemical point of view. In my opinion, the study group should comprise people with confirmed laboratory indicators of insulin resistance, e.g. based on HOMA-IR.

We thank the reviewer for raising this important point and we would happily address it. Including overweight/obese subjects regardless of their insulin resistance status for assessing the acute insulin-sensitizing effects of exercise is common practice in the literature. And this inclusion criteria sometimes leads to the inclusion in the same study of overweight/obese participants that do and do not present clinically established insulin resistance [3–5], or even do and do not present impaired fasting glycemia [6], to cite a few of many studies. Furthermore, it is an incomplete understanding of our inclusion criteria considering we would include participants only by assessing body mass index (although again, this is common practice in the relevant literature), as we clearly state that central obesity, assessed by waist circumference, is also to be used as a screening tool. Nevertheless, to make this point even more clear, and to add 2 more screening tools to characterize obesity and central adiposity, we added the following sentence under “Description of participants”:

“In order to confirm the obesity and central adiposity statuses of the participants, dual-energy X-ray absorptiometry will be used to assess fat percentage and visceral fat mass (see details below in Anthropometric measurements), and participants will be included only if both indexes are >90th percentile for age and sex [37].”

Also, many studies were cited throughout the paper to show that within the type 2 diabetes mellitus spectrum, and regardless of the disease status (from healthy to obesity to insulin resistance to prediabetes to type 2 diabetes mellitus) the effects of resistance exercise on improving glucose metabolism have been observed.

Finally, we acknowledge this potential source of variation in the “Weakness” section within the “Discussion” chapter:

“Furthermore, individuals with potentially different levels of insulin resistance might be included in this study, and greater exercise-induced improvement in insulin resistance has been observed in individuals with higher levels of baseline insulin resistance [28,102]. Thus, this factor can also increase results variability. However, it should be pointed out that studying overweight/obese subjects regardless of their insulin resistance or fasting glucose impairment status is commonplace in the related literature [8,103–105]. Nevertheless, the fact that participants will be their own control minimizes this potential source of variability. Moreover, individual results will be presented and discussed accordingly should individual levels of insulin resistance affect average group results. Also, DM2 patients will be excluded, following our exclusion criteria.”

Therefore, we would like to keep our inclusion criteria as it stands, and hope the above clarification helps on settling this issue.

– Measuring insulin resistance based on the OGTT result seems to represent imprecise methodology. Basing basically the entire study on this parameter seems to be an oversimplification and lead to false conclusions.

We thank the reviewer for addressing this point. We expanded on the explanation as to why will be used, based on the relevant literature under “Strengths” within the “Discussion” chapter:

“Finally, although the OGTT might not be the gold-standard tool to assess insulin sensitivity compared to the euglycemic-hyperinsulinemic clamp (EHC), ingesting 75 grams of glucose is considered to be a more physiological stimulus compared the supraphysiological insulin levels during EHC [90]. Moreover, the insulin sensitivity indexes that will be calculated from OGTT in the present study show high correlation (r=0.61 and 0.96) [91] with EHC results. Also, OGTT is considered reliable and consistent for estimating insulin sensitivity over consecutive days [92]. Last, but not the least, the majority of studies that assessed the acute effects of a strength exercise session on insulin sensitivity employed OGTT as the assessment tool [28,50,53,93–101].”

Thus, we respectfully disagree with impression the reviewer has on the OGTT being imprecise and that its results can lead to false conclusions, as the literature presented in the sentence above shows the exact opposite, and majority of studies on the effects of resistance exercise on glucose metabolism were performed using this technique.

– Performing a glucose tolerance test several hours after exercise seems inappropriate, because "acute" processes related to glucose utilization by the tissues are still taking place at that time. Thus, the determined parameters will reflect the body's acute response to increased glucose utilization rather than the possible increase in insulin sensitivity. It seems reasonable to increase the time interval between the training session and blood drawing and measure the true change in insulin sensitivity,

We thank the reviewer for addressing this point, and would like to make it clear that the present protocol was designed to investigate the acute (one bout), not chronic (training, several bouts) effects of resistance exercise. This is based on the evidence that the exercise training effects on improved insulin sensitivity are not a true training effect, but the lingering acute effect of the last exercise session [5,7], which suggests that understanding the acute effects of exercise on insulin sensitivity is of great clinical importance. Therefore, the proximity between the exercise bout and the OGTT is by design, not a methodological limitation.

– Small number of participants in the study. Such sample size does not ensure that analyzed parameters will reach adequate power. In the power analysis of the test only the Matsuda insulin sensitivity index was used.

To address this potential anticipated limitation, we had previously calculated sample size based on all insulin sensitivity indexes we will assess (i.e., the oral glucose insulin sensitivity index [8], the Matsuda insulin sensitivity index [9] and Cederholm's index [10], muscle insulin sensitivity index [11,12], glucose-stimulated insulin sensitivity index [13], oral disposition index [14,15], Gutt index [16], Avignon et al. [17], Belfiore et al. index [18], Stumvoll et al. index [19], and McAuley et al. index [20]), and when we used the Matsuda index the number of participants needed was the greatest. That is why we chose this index tom calculate sample size. Had we based the calculation exclusively on another index, sample size would have been smaller, and the chance of the study being underpowered for the other indexes would increase. Finally, an inspection of the sample size from the literature pertaining to the effects of resistance exercise on insulin sensitivity [1,21–36] reveals an average of 15.95 participants, with a huge variation (SD = 8.23). Therefore, we are confident that we were diligent when calculating our sample size, as we used the index that gave us the greatest number of subjects needed, and the calculated sample size is within the sample sizes used in studies that addressed similar research problems.

– Presented research program is not innovative.

Once again, we respectfully disagree with the impression of the reviewer. In a thorough literature review, we were unable to find one single study that had similar study design, especially when considering the scientific rigor we are employing in our protocol (adhering the Spirit guidelines, blinding of data analyst, blinding of statistician, etc.). If the reviewer is aware of such a study from which the current protocol does not differ, or innovates from, we kindly ask for the reference, and we would gladly read this paper, and discuss or even change our protocol accordingly.

The subject matter of the work is more consistent with journals on physiotherapy or exercise physiology.

In the PlosOne website, under the scope of the journal, one reads:

“PLOS ONE welcomes original research submissions from the natural sciences, medical research, engineering, as well as the related social sciences and humanities, including: 

Protocols, including Lab Protocols that describe verified methodologies and Study Protocols that describe detailed plans for research projects.

[authors underlined the text above].

Therefore, we the authors were somewhat surprised with the above statement by the reviewer, since at no moment PlosOne guidelines prohibit or exclude papers from one particular area. Quite the opposite: it welcomes papers from medical research and study protocols. In fact, a quick search in the PlosOne website reveals many study protocols that relate to similar medical areas [37,38] the current protocol falls into. Thus, we do not believe a paper in PlosOne should be rejected or even criticized based on the area it more closely relates to, especially if the paper falls within the scope of the journal, and previous similar papers were published recently in the journal.

References

1. Tong TK, Kong Z, Shi X, Shi Q. Comparable Effects of Brief Resistance Exercise and Isotime Sprint Interval Exercise on Glucose Homeostasis in Men. J Diabetes Res. 2017;2017. doi:10.1155/2017/8083738

2. Kahn M, Korhonen T, Leinonen L, Martinmaki K, Kuula L, Pesonen A-K, et al. Is It Time We Stop Discouraging Evening Physical Activity? New Real-World Evidence From 150,000 Nights. Front Public Health. 2021;9. doi:10.3389/fpubh.2021.772376

3. Whyte LJ, Ferguson C, Wilson J, Scott RA, Gill JMR. Effects of single bout of very high-intensity exercise on metabolic health biomarkers in overweight/obese sedentary men. Metabolism. 2013;62: 212–219. doi:10.1016/j.metabol.2012.07.019

4. Durrer C, Robinson E, Wan Z, Martinez N, Hummel ML, Jenkins NT, et al. Differential Impact of Acute High-Intensity Exercise on Circulating Endothelial Microparticles and Insulin Resistance between Overweight/Obese Males and Females. PLoS One. 2015;10: e0115860. doi:10.1371/journal.pone.0115860

5. Ryan BJ, Schleh MW, Ahn C, Ludzki AC, Gillen JB, Varshney P, et al. Moderate-Intensity Exercise and High-Intensity Interval Training Affect Insulin Sensitivity Similarly in Obese Adults. J Clin Endocrinol Metab. 2020;105: e2941–e2959. doi:10.1210/clinem/dgaa345

6. Little JP, Jung ME, Wright AE, Wright W, Manders RJF. Effects of high-intensi

---

## [Decision Letter · Decision Letter 1]

5 Mar 2024

PONE-D-23-04520R1The ASSIST trial: Acute effectS of manipulating Strength exercise volume on Insulin SensiTivity in obese adults: a protocol for a randomized controlled, crossover, clinical trialPLOS ONE

Dear Dr. Magalhaes,

Thank you for submitting your manuscript to PLOS ONE. After careful consideration, we feel that it has merit but does not fully meet PLOS ONE’s publication criteria as it currently stands. Therefore, we invite you to submit a revised version of the manuscript that addresses the points raised during the review process.

Please go over the comments reported by one of the reviewers (bottom of this email) and submit your revised manuscript in March 26th, 2024. If you will need more time than this to complete your revisions, please reply to this message or contact the journal office at plosone@plos.org. Please include the following items when submitting your revised manuscript:A rebuttal letter that responds to each point raised by the academic editor and reviewer(s). You should upload this letter as a separate file labeled 'Response to Reviewers'.A marked-up copy of your manuscript that highlights changes made to the original version. You should upload this as a separate file labeled 'Revised Manuscript with Track Changes'.An unmarked version of your revised paper without tracked changes. You should upload this as a separate file labeled 'Manuscript'.If applicable, we recommend that you deposit your laboratory protocols in protocols.io to enhance the reproducibility of your results. Protocols.io assigns your protocol its own identifier (DOI) so that it can be cited independently in the future. For instructions see: https://journals.plos.org/plosone/s/submission-guidelines#loc-laboratory-protocols. Additionally, PLOS ONE offers an option for publishing peer-reviewed Lab Protocol articles, which describe protocols hosted on protocols.io. Read more information on sharing protocols at https://plos.org/protocols?utm_medium=editorial-email&utm_source=authorletters&utm_campaign=protocols.

We look forward to receiving your revised manuscript.

Kind regards,

Everson Nunes, Ph.D.

Academic Editor

PLOS ONE

Journal Requirements:

"NO - The funders did not and will not have a role in study design, data collection and analysis, decision to publish, or preparation of the manuscript."

Reviewers' comments:

Reviewer's Responses to Questions

**Comments to the Author**

1. Does the manuscript provide a valid rationale for the proposed study, with clearly identified and justified research questions?

Reviewer #2: Yes

Reviewer #3: Partly

2. Is the protocol technically sound and planned in a manner that will lead to a meaningful outcome and allow testing the stated hypotheses?

Reviewer #2: Yes

Reviewer #3: Partly

3. Is the methodology feasible and described in sufficient detail to allow the work to be replicable?

Reviewer #2: Yes

Reviewer #3: Yes

4. Have the authors described where all data underlying the findings will be made available when the study is complete?

Reviewer #2: Yes

Reviewer #3: Yes

5. Is the manuscript presented in an intelligible fashion and written in standard English?

Reviewer #2: Yes

Reviewer #3: Yes

6. Review Comments to the Author

You may also provide optional suggestions and comments to authors that they might find helpful in planning their study.

Reviewer #2: I have no comments, the authors made necessary amendments and clarified doubtful issues. This study may be valuable contribution to our understanding of insulin sensitivity.

Reviewer #3: Great job on this study design, below are a few comments:

- personally I would suggest adjusting the title and making it more specific. Make sure it is clear that the study is focusing on an acute period, not long term. Stating a questions like this is too general and does not represent what you are investigating.

- Currently there is a large difference in exercise volume. Consider adding one group that does 2 sets so you have one group that does 14 sets. This allows you to make a firmer statement.

- It is not fully clear to me what this study will ad on top of the already available studies. Black et al Is mentioned in the manuscript and I am not convinced that the current study will add a lot of information on to this paper. It looks to me as a study designed to confirm what is already been done.

- The cross over model is great!

- It is important to keep the time in between days similar within and between the participants. A range of 4 to 28 days between trials is now given which to me seems like a lot of variance.

- The 8RM for all 7 exercises in 1 day seems like a lot. How are you going to ensure that the participant is not fatigued at the last exercise? 2 minutes of rest is stated, is this sufficient?

- It is suggested to base your exclusion criteria on baseline insulin resistance. Personally I would suggest picking some cut off values to make the study more specific and to prevent outliers in your data.

- A lot of details are giving in the explanation of the exercise and diet for all the conditions. This part seems well thought out to me. Good job.

- I would suggest going over the structure of some sentences again to make sure there is a nice flow in the manuscript.

7. PLOS authors have the option to publish the peer review history of their article (what does this mean?). If published, this will include your full peer review and any attached files.

Reviewer #2: No

Reviewer #3: No

---

## [Author Response · Author response to Decision Letter 1]

7 Mar 2024

Editor comments.

We thank the editor for giving us the opportunity to reply to the reviewers’ comments. Our replies are in bold following each comment.

Journal Requirements: 

We changed the format of sections to match the journal’s requirements.

2. We note that the grant information you provided in the ‘Funding Information’ and ‘Financial Disclosure’ sections do not match.When you resubmit, please ensure that you provide the correct grant numbers for the awards you received for your study in the ‘Funding Information’ section.

We doubled edited the funding information in the main text to match the one provided in the submission platform. Also, we removed the institution support (basically equipment and space) to the acknowledgments section, as the institutional where data were collected did not provide financial support.

"NO - The funders did not and will not have a role in study design, data collection and analysis, decision to publish, or preparation of the manuscript."

We changed the text accordingly in the “Funding and role of funding source” section to:

This study is supported by the Conselho Nacional de Desenvolvimento Científico e Tecnológico (CNPQ: Grant#407975/2018-7 and #402091/2021-3) and by the Fundação de Amparo à Pesquisa do Estado de Minas Gerais (FAPEMIG: Grant#APQ-00008-22). The funders had no role in study design, data collection and analysis, decision to publish, or preparation of the manuscript.

We changed the text in the submission platform and in S2_Table to:

“Deidentified research data will be made publicly available when the study is completed and published.”

We are not aware of any citation included that was retracted.

 

Reviewers' comments:

Reviewer #2: I have no comments, the authors made necessary amendments and clarified doubtful issues. This study may be valuable contribution to our understanding of insulin sensitivity.

We appreciate the compliment, and thank the reviewer for their time and effort in reviewing this study. 

 

Reviewer #3: Great job on this study design, below are a few comments:

We appreciate the compliment, and thank the reviewer for their time and effort in reviewing this study. 

- personally I would suggest adjusting the title and making it more specific. Make sure it is clear that the study is focusing on an acute period, not long term. Stating a questions like this is too general and does not represent what you are investigating.

We thank the reviewer for this comment. In fact, the term “acute” is the very first word in the title, so respectfully, we do not see the need to restate that the study is acute in the title once more. Should the reviewer have a suggestion for changing the title, we can certainly change it accordingly.

- Currently there is a large difference in exercise volume. Consider adding one group that does 2 sets so you have one group that does 14 sets. This allows you to make a firmer statement.

This large difference in exercise volume is intentional. Although we agree that a dose-response study on the effect of resistance exercise on insulin sensitivity is warranted, our rationale for the present study relates to time-commitment, as stated in the introduction. As sedentary, obese individuals might feel demotivated to perform a traditional, high-volume resistance exercise session due to time constraints and low self-efficacy, our idea was to drastically reduce exercise volume, and by default, session duration, in order to assess whether a shorter session, with lower volume would still be effective in promoting beneficial effects on insulin sensitivity. Thus, our aims in this stufy do not relate to assessing a dose-response relationship. On top of that, another situation with 14 sets would increase the burden on the subjects, potentially reducing participation adherence. 

To address this important point raised by the reviewer, we included the following sentence in the “weakness” subsection within the Discussion section:

“Finally, per our study design, we will assess two significantly different strength exercise volumes, and by default time commitment, as lack of time is usually listed as one of the main reasons not to adhere to exercise [31]. Thus, our study design is not intended to explore a potential dose-response relationship between strength exercise volume and improvements in insulin sensitivity, although this approach would bear practical implications.”

- It is not fully clear to me ppwhat this study will ad on top of the already available studies. Black et al Is mentioned in the manuscript and I am not convinced that the current study will add a lot of information on to this paper. It looks to me as a study designed to confirm what is already been done.

We thank the reviewer for the opportunity to make the case for our study design and outline how it differs from previous literature, specially form Black et al [1]. There are many points that the current study design differs, and in our opinion, improves on the Black’s et al’s study. First, authors assessed women with childbearing capacity, but did not report, nor discussed menstrual cycle phase, different from our study design. Second, they only performed one familiarization session, which was followed by a 5-RM strength test session. Having previously strength exercise naïve subjects perform only one familiarization session before engaging in a quite demanding strength test session is not best practice, as true familiarization most likely was not attained. Not being fully familiarized with the resistance exercises is a big concern when assessing insulin sensitivity, as unaccustomed strength exercise can lead to worsening of insulin sensitivity [2–4]. Our current study design anticipates at least 4 familiarization sessions before the strength tests session to ensure participants are fully familiarized with the research protocol. Third, the study of Black et al used the glucose and insulin data from the morning before the exercise sessions as baseline, when best practice on clinical trials dictates a control (sham) day should be employed, as will be done in the present study. Fourth, that study assessed insulin sensitivity only by calculating HOMA-IR. Not only HOMA-IR is a much more limited tool to address insulin sensitivity compared to OGTT-derived indexes[5], but it is actually a measure of insulin resistance, more specifically, liver insulin resistance[6], and insulin sensitivity is not the mere opposite of insulin resistance [7]. In the present study, we will assess HOMA-IR, but most importantly, OGTT-derived indexes will provide us the opportunity to tease out changes in muscle insulin sensitivity and liver insulin resistance as they change in response to a strength exercise bout. Fifth, Black et al do not report any actions taken toward blinding glucose and insulin measurements, nor the statistical analysis, different from what we propose to do. Sixth, and perhaps most importantly, Black et al do not describe the degree of effort (or proximity to failure) in each set. This is very important because as stated in our introduction and discussion, the degree of effort is regarded as a factor that affects the effect on strength exercise on glucose and insulin responses [8]. Although degree of effort was not reported, per Black et al’s design, it can be assumed that in the “high-volume” situations the degree of effort was higher, as fatigue would have accumulated with multiple sets. Our design anticipates participants will perform all sets in both exercise sessions to failure, thus a true volume effect will be isolated from this important confounding variable.

Altogether, we believe the current design of our study improves on the design reported by Black et al and we believe the data generated by our study will stem from a much higher methodological quality. 

To address Black et al’s limitations, we added the following sentence:

“Black et al [29] evaluated 8 exercises performed with 1 or 4 sets, either at 65% 1RM (12-15 reps) or 85% 1RM (6 to 8 reps). The authors observed improvements in insulin resistance in all protocols; but, protocols with single sets showed less effect. However, in that study, the degree of effort in each set was not reported and arguably higher in the high-volume conditions, and as this parameter is suggested as a determinant in the improvement of insulin sensitivity [25], one cannot exclude the possibility this factor played a role in the results observed. Furthermore, Black et al [29] assessed insulin resistance by the homeostasis model assessment of insulin resistance (HOMA-IR), which is more limited than other methods, such as OGTT-derived indexes [30], and more closely reflects hepatic insulin resistance [31].”

- The cross over model is great!

We appreciate the compliment.

- It is important to keep the time in between days similar within and between the participants. A range of 4 to 28 days between trials is now given which to me seems like a lot of variance.

As stated in the “Study type and design” section, if the interval between sessions are to be more than 21 days apart, participants will perform another familiarization session 1 to 2 weeks preceding the following session, thus the interval between a familiarization and a study session will be 4 to 14 days. We had to do this, because as women with childbearing potential can be enrolled, and we aim to assess them in the follicular phase of the menstrual cycle, we had to figure out a way to keep them familiarized to the study protocol, but at the same avoiding a training effect. It has been shown that even once a week strength training can lead to improvement in strength [9,10]. So, we could not have them familiarized once a week (nor more frequent than that), because this could lead to a training effect. To address this point, we included the following statement to “Study type and design”:

“The interval between experimental sessions will be recorded and reported in the final manuscript, and discussed accordingly.”

- The 8RM for all 7 exercises in 1 day seems like a lot. How are you going to ensure that the participant is not fatigued at the last exercise? 2 minutes of rest is stated, is this sufficient?

Performing strength tests in all seven exercises in the same strength session is intentional. As participants will perform the seven exercises in the exercise sessions back-to-back, we wanted them to be tested in the same fashion. Although this design will probably lead to some fatigue carryover, we believe it more closely aligns with the practical applications of our study, and mimics the actual fatigue participants will have to endure during the exercise sessions. Also, this is not a training study, and the measurement of strength in the present protocol will only serve to prescribe the load during the exercise sessions. Finally, in both exercise sessions, they will lift the exact same load, so even if we do not find a “real” 8-RM in every exercise, this will have no impact our study design or results’ interpretation.

To address this, we added the following statement to the “Strength tests” subsection:

“Because during the high- and low-volume experimental sessions participants will perform all seven exercises sequentially, and fatigue will likely accumulate as the sessions progresses, we decided to have their 8-RM tested in the same fashion, in order to mimic their anticipated effort. This means a true 8-RM might not be recorded, especially for exercises tested later in the strength test session. However, we believe this will not interfere with our study design and results interpretation, as exactly the same load will be prescribed for the exercise experimental sessions.”

- It is suggested to base your exclusion criteria on baseline insulin resistance. Personally I would suggest picking some cut off values to make the study more specific and to prevent outliers in your data.

Respectfully, we believe we have already addressed this point in the “Weaknesses” subsection within the discussion section, as it reads below:

“Furthermore, individuals with potentially different levels of insulin resistance might be included in this study, and greater exercise-induced improvement in insulin resistance has been observed in individuals with higher levels of baseline insulin resistance [29,102]. Thus, this factor can also increase results variability. However, it should be pointed out that studying overweight/obese subjects regardless of their insulin resistance or fasting glucose impairment status is commonplace in the related literature [8,103–105]. Nevertheless, the fact that participants will be their own control minimizes this potential source of variability. Moreover, individual results will be presented and discussed accordingly should individual levels of insulin resistance affect average group results. Also, DM2 patients will be excluded, following our exclusion criteria.”

- A lot of details are giving in the explanation of the exercise and diet for all the conditions. This part seems well thought out to me. Good job.

We appreciate the compliment.

- I would suggest going over the structure of some sentences again to make sure there is a nice flow in the manuscript.

We appreciate this comment, and have performed a thorough revision in the entire manuscript, with special attention to the introduction, as now in hindsight we believe was longer than necessary. The specific changes can be tracked in the track changes version.

 

References

1. Black

---

## [Decision Letter · Decision Letter 2]

4 Apr 2024

The ASSIST trial: Acute effect s  of manipulating  s trength exercise volume on  i nsulin  s ensi t ivity in obese adults: A protocol for a randomized controlled, crossover, clinical trial

PONE-D-23-04520R2

Dear Dr. Magalhaes, 

We’re pleased to inform you that your manuscript has been judged scientifically suitable for publication and will be formally accepted for publication once it meets all outstanding technical requirements.

Kind regards,

Everson Nunes, Ph.D.

Academic Editor

PLOS ONE

Additional Editor Comments (optional):

Reviewers' comments:

Reviewer's Responses to Questions

**Comments to the Author**

1. Does the manuscript provide a valid rationale for the proposed study, with clearly identified and justified research questions?

Reviewer #2: Yes

Reviewer #3: Yes

2. Is the protocol technically sound and planned in a manner that will lead to a meaningful outcome and allow testing the stated hypotheses?

Reviewer #2: Yes

Reviewer #3: Yes

3. Is the methodology feasible and described in sufficient detail to allow the work to be replicable?

Reviewer #2: Yes

Reviewer #3: Yes

4. Have the authors described where all data underlying the findings will be made available when the study is complete?

Reviewer #2: Yes

Reviewer #3: Yes

5. Is the manuscript presented in an intelligible fashion and written in standard English?

Reviewer #2: Yes

Reviewer #3: Yes

6. Review Comments to the Author

You may also provide optional suggestions and comments to authors that they might find helpful in planning their study.

Reviewer #2: I have no comments, the authors made necessary amendments and clarified doubtful issues. This study may be valuable contribution to our understanding of insulin sensitivity.

Reviewer #3: In my opinion, the authors have made the necessary changes to their manuscript to be accepted.

With regard to my previous comment: "It is suggested to base your exclusion criteria on baseline insulin resistance.

Personally I would suggest picking some cut off values to make the study more specific

and to prevent outliers in your data."

Even though this is mentioned in the weaknesses part, I would strongly recommend changing the inclusion/exclusion criteria. Simply mentioning it as a weakness does not make it justifiable.

7. PLOS authors have the option to publish the peer review history of their article (what does this mean?). If published, this will include your full peer review and any attached files.

Reviewer #2: No

Reviewer #3: No

---

## [Editor Report · Acceptance letter]

29 Apr 2024

PONE-D-23-04520R2 

PLOS ONE

Dear Dr. de Castro Magalhaes, 

I'm pleased to inform you that your manuscript has been deemed suitable for publication in PLOS ONE. Congratulations! Your manuscript is now being handed over to our production team.

Kind regards, 

on behalf of

Dr. Everson Nunes 

Academic Editor

PLOS ONE